# Expression Profiles of ID and E2A in Ovarian Cancer and Suppression of Ovarian Cancer by the E2A Isoform E47

**DOI:** 10.3390/cancers14122903

**Published:** 2022-06-12

**Authors:** Yong-Jae Lee, Eun-Ji Nam, Sunghoon Kim, Young-Tae Kim, Pamela Itkin-Ansari, Sang-Wun Kim

**Affiliations:** 1Department of Obstetrics and Gynecology, Institute of Women’s Medical Life Science, Yonsei University College of Medicine, Seoul 03722, Korea; svass@yuhs.ac (Y.-J.L.); nahmej6@yuhs.ac (E.-J.N.); shkim70@yuhs.ac (S.K.); ytkchoi@yuhs.ac (Y.-T.K.); 2Development, Aging and Regeneration Program, Sanford Burnham Prebys Medical Discovery Institute, La Jolla, CA 92037, USA

**Keywords:** ovarian cancer, basic helix-loop-helix protein, inhibitor of DNA binding, transcription factor-3, bHLH/ID complex

## Abstract

**Simple Summary:**

Ovarian cancer is one of the most fatal gynecological malignancies in women. Even though primary treatment might result in complete remission, approximately 60–80% patients with advanced-stage ovarian cancer experience a relapse. To improve their survival outcomes, several novel agents have been proposed; however, the majority of these have demonstrated limited efficacy in the treatment of ovarian cancer. Interestingly, an imbalance in the basic helix-loop-helix inhibitor of DNA binding (bHLH/ID) protein networks has been associated with oncogenesis, but the exact expression patterns of bHLH/ID in ovarian cancer are not yet known. We hypothesized that such an imbalance in bHLH/ID activity may be integral to ovarian cancer pathogenesis, and modulating the bHLH/ID balance might be a promising approach to treat ovarian cancer.

**Abstract:**

The E2A and inhibitor of DNA binding (ID) proteins are transcription factors involved in cell cycle regulation and cellular differentiation. Imbalance of ID/E2A activity is associated with oncogenesis in various tumors, but their expression patterns and prognostic values are still unknown. We evaluated ID and E2A expression in ovarian cancer cells, and assessed the possibility of reprogramming ovarian cellular homeostasis by restoring the ID/E2A axis. We analyzed copy number alterations, mutations, methylations, and mRNA expressions of ID 1–4 and E2A using The Cancer Genome Atlas data of 570 ovarian serous cystadenocarcinoma patients. Incidentally, 97.2% cases exhibited gain of ID 1–4 or loss of E2A. Predominantly, ID 1–4 were hypomethylated, while *E2A* was hypermethylated. Immunohistochemical analysis revealed that ID-3 and ID-4 expressions were high while E2A expression was low in cancerous ovarian tissues. Correlation analysis of ID and E2A levels with survival outcomes of ovarian cancer patients indicated that patients with high ID-3 levels had poor overall survival. We also determined the effect of E2A induction on ovarian cancer cell growth in vitro and in vivo using SKOV-3/Luc cells transduced with tamoxifen-inducible E47, a splice variant of E2A. Interestingly, E47 induced SKOV-3 cell death in vitro and inhibited tumor growth in SKOV-3 implanted mice. Therefore, restoring ID/E2A balance is a promising approach for treating ovarian cancer.

## 1. Introduction

Basic helix-loop-helix (bHLH) proteins are transcription factors that bind to cognate E-box sequences and induce the expression of specific genes, including those that promote differentiation of stem cells or progenitor cells into specific cell types. Moreover, they play key roles in maintaining the differentiated cell state and inhibiting cell proliferation. Interestingly, the bHLH family members assemble as homodimer or heterodimer complexes with other bHLH proteins [1].

The transcription factor-3 gene (TCF-3), which is the founding member of the E-protein gene family, encodes the E2A proteins. In fact, E2A encodes two distinct bHLH transcription factors, E12 and E47, produced by alternative splicing and collectively known as the E2A proteins. These E2A proteins regulate the expression of genes that control cell growth and differentiation in various cell lineages [2]. Incidentally, E2A proteins are negatively regulated by the inhibitor of DNA binding (ID) family of proteins, particularly ID 1–4 in humans. These ID proteins contain an α-helical HLH dimerization motif, but they lack a DNA-binding domain; hence, bHLH/ID heterodimers are dominant-negative inhibitors of bHLH [3]. Therefore, ID proteins can control cell differentiation by interfering with the DNA-binding activity of E2A proteins, which, in turn, makes them the master regulators of cancer stem cells and tumor aggression. Recent studies have shown that ID genes may function as oncogenes, and ID proteins might be inhibitors of G1 cell cycle arrest and cell differentiation [4,5]. In fact, overexpression of ID proteins has been demonstrated in various cancers [6,7]. Reportedly, ID genes promote cell cycle progression, and their overexpression induces apoptosis as well as oncogenesis [8]. Additionally, ID proteins are necessary for angiogenesis and vascularization of tumors [9]. In breast cancer, high expression levels of ID proteins are associated with cancer proliferation and metastasis [10,11,12]. Moreover, overexpression of ID3 has been observed in cases of esophageal squamous cell carcinoma [13] and cervical cancer [14]. According to a previous study, ID proteins are highly expressed in pancreatic ductal adenocarcinoma (PDA); therefore, increasing E2A expression in such patients can cause growth arrest and cellular re-differentiation, thereby indicating that E2A is a primary target of ID proteins in cancer [15]. Additionally, the E2A expression is lower in colorectal cancer tissue than in normal colon mucosa. In fact, in highly advanced colon cancer patients, the lower the expression level of E2A, the poorer the prognosis. Furthermore, previous studies indicate that in colon cancer patients, cancer cell proliferation is accelerated when E2A expression is suppressed and is inhibited when E2A is overexpressed [16,17].

Ovarian cancer is a highly fatal gynecological malignancy in women [18]. Optimal cytoreductive surgery combined with platinum-based chemotherapy is the standard treatment for advanced-stage ovarian cancer [19]. However, even in cases of complete remission post-primary treatment, approximately 60–80% of these patients experience relapse [20]. To improve survival outcomes, several new anti-cancer drugs have been developed, particularly for targeted therapy, anti-angiogenesis, and immunological modification [12]. Incidentally, ID proteins have been considered as possible targets for novel therapeutic agents. In fact, the level of ID-1 expression correlates with the malignant potential of ovarian cancer, resulting in poor survival outcomes [12]. Reportedly, ID protein inhibition by a peptide aptamer (ID1/3-PA7) induces cell cycle arrest and apoptosis in ovarian cancer [21]. Incidentally, the ID*4* gene is amplified in ovarian cancer; moreover, an ID4-specific tumor-penetrating nano-complex can suppress cancer growth and significantly improve the survival of tumor-bearing mice [22]. However, a precise understanding of ovarian cancer with respect to the expression levels of ID and E2A is necessary to improve the efficiency of therapeutic concepts.

We hypothesized that dysregulated bHLH activity may be integral to ovarian cancer pathogenesis. Additionally, an increased E2A expression might restore the level of E2A activity that is necessary to induce ovarian cancer cell growth arrest or apoptosis.

In this study, we analyzed ovarian cancer samples from The Cancer Genome Atlas (TCGA) database to investigate the degree of mutation or differential expression of ID 1–4 and E2A genes as well as the role of E47 as a growth suppressor in ovarian cancer cells. We also examined ID and E2A expression in tumor tissues of ovarian cancer patients and determined the significance of overexpressing the E2A splice variant E47 in ovarian cancer.

## 2. Materials and Methods

### 2.1. TCGA Data

We analyzed the data of 570 patients with ovarian serous cystadenocarcinoma from TCGA (cBioPortal for Cancer Genomics, UCSC cancer genomics browser) database. Particularly, the copy number variations, methylation status, and mRNA expression levels of ID 1–4 and E2A genes were evaluated at different stages of cancer. Additionally, we analyzed the associations between copy number alterations of the genes and their corresponding mRNA expression levels, as well as differential mRNA expression and degrees of gene methylation.

### 2.2. Cell Culture

The human ovarian cancer cell line SKOV-3 was obtained from the American Type Culture Collection (Manassas, VA, USA), and the cells were cultured in RPMI 1640 (Gibco) medium containing 10% fetal bovine serum in a humid atmosphere with 5% CO_2_ at 37 °C. Subsequently, the SKOV-3 cells were infected with a retroviral vector expressing E47 fused to a tamoxifen-inducible modified estrogen receptor (MER) to generate E47-inducible cells (SKOV-3/E47^MER^). This vector also contained the interleukin 2 receptor alpha gene (*IL2RA*) that encodes CD25; therefore, CD25-expressing cells were selected to isolate the stable cell lines using fluorescence-activated cell sorting (FACS), as previously described [15,23]. The E47 activity was induced by incubating the cells with 4 μmol/L tamoxifen (Sigma-Aldrich, St. Louis, MO, USA) for 48 h, unless otherwise noted.

### 2.3. Transfection, Cell Cycle Analysis, and FACS

For flow cytometry analysis, the SKOV-3 cells were transfected (Lipofectamine 2000; Invitrogen) with green fluorescent protein (SKOV-3/Luc) or inducible E47 plasmids (SKOV-3/Luc/E47^MER^) and incubated with tamoxifen for 48–72 h. For live cell sorting, the SKOV-3 cells were immunostained with fluorescein isothiocyanate (FITC)-conjugated mouse anti-human CD25 (1:100, BD Biosciences, San Jose, USA), as previously described [24]. For cell cycle analysis, the cells were fixed with 100% ethanol, incubated with anti-human CD25 and propidium iodide (PI; Invitrogen Grand Island, NY, USA), and analyzed using a FACS Canto cytometer (BD Biosciences, San Jose, USA). The G0/G1, S, and G2/M phase estimates were generated by modeling data with ModFitLT software (Verity Software House, Topsham, ME, USA).

### 2.4. Apoptosis Analysis

For detection of cell apoptosis, the annexin V- FITC/PI apoptosis detection kit (Apoptosis Detection Kit II; BD Pharmingen, Franklin Lakes, NJ, USA) was used, according to the manufacturer’s protocol. First, the cells and supernatant were collected, washed twice with cold 1× phosphate-buffered saline (PBS), and resuspended in 1× binding buffer. Thereafter, 5 μL of annexin V-FITC and PI was added to the solution, followed by incubation in the dark for 15 min at 37 °C. Finally, stained cells were detected using FACS (BD Biosciences).

### 2.5. Microarray Analysis

Cells were harvested from the group incubated with tamoxifen for 48 h (SKOV-3/Luc/E47^MER^), and from the untreated group of cells (SKOV-3/Luc). For the microarray analysis, the RNA was first labeled with biotin-16-UTP and subsequently hybridized to the HumanHT-12 v4 Expression BeadChip (Illumina, Inc., San Diego, CA, USA). Thereafter, the BeadChip was scanned and normalized using a BeadArray Reader. The resulting data were collected by scanner software and pre-processed using GenomeStudio software (Illumina, Inc.). Subsequently, principal component analysis of differential gene expression was performed using the Partek Genomics Suite software (Partek, Inc., St. Louis, MO, USA). Hierarchical clustering and other statistical analyses were performed using the Database for Annotation, Visualization, and Integrated Discovery (DAVID; https://david.ncifcrf.gov/summary.jsp, accessed on 3 November 2019). Finally, the functions of *E47* and the genes significantly associated with *E47* alterations were predicted by Gene Ontology (GO) and Kyoto Encyclopedia of Genes and Genomes (KEGG) pathway enrichment analysis of the differentially expressed genes obtained from the DAVID.

### 2.6. Real-Time Quantitative Polymerase Chain Reaction (qPCR) Analysis

The ID and E2A expression levels in ovarian tissues of patients with ovarian cancer (*n* = 98) and normal ovarian tissues (*n* = 20) were analyzed by real-time qPCR. Total RNA was extracted from the ovarian tissue samples using the RNeasy Mini Kit (Qiagen, Germantown, MD, USA) and subjected to reverse transcription with qScript cDNA Supermix (Quanta, Gaithersburg, MD, USA), according to the manufacturer’s instructions. Ultimately, real-time qPCR was performed using the LightCycler 480 II system with SYBR Green I (Roche, Basel, Switzerland), and the gene expression level was normalized to 18S rRNA.

### 2.7. Animal Studies

We obtained 10 severe combined immunodeficiency (SCID)/beige mice from Orient Bio Co. Ltd. (Seoul, Korea). Prior to implantation, the SKOV-3/Luc and SKOV-3/Luc/E47^MER^ cells were treated with tamoxifen (4 uM), and the cell proliferation changes were observed. Subsequently, 3 million live cells of SKOV-3/Luc and SKOV-3/Luc/E47^MER^ were subcutaneously implanted into the SCID/beige mice. After three days of implantation, tamoxifen (free base) pellets (25 mg/pellet, 90 day release; Innovative Research of America, Sarasota, FL, USA) were implanted subcutaneously at other sites to analyze the effect of E47 on tumorigenesis and growth of SKOV-3/Luc cells. Tumor size was measured weekly using ultrasound imaging. Moreover, the Xenogen IVIS imaging system was used for bioluminescence imaging, in which the luminescence was measured after an intraperitoneal injection of luciferin solution (15 mg/mL or 30 mg/kg in PBS; dose of 150 mg/kg) every 2 min between 5 and 20 min, and the maximum value was used for further analysis. The mice were humanely killed when they showed clinical signs of distress or pain, including hunched posture, ruffled coat, open sores, slow breathing, or reduced response to external stimuli. The mice were euthanized by CO_2_ asphyxiation.

### 2.8. Immunohistochemistry (IHC) Analysis

Formalin-fixed paraffin-embedded (FFPE) tissues were stained using the Ventana BenchMark XT automated immunostainer (Ventana Medical Systems, Tucson, AZ, USA), according to the manufacturer’s instructions. Thereafter, the slides were dried at 60 °C for 1 h and deparaffinized using EZ Prep (Ventana Medical Systems, Tucson, AZ, USA) at 75 °C for 4 min. Cell conditioning was performed using the CC1 solution (Ventana Medical Systems, Tucson, AZ, USA) at 100 °C for 64 min. Subsequently, the slides were incubated with ID-1, ID-2, ID-3, and ID-4 rabbit polyclonal antibodies (Abcam, Cambridge, UK) and E2A rabbit polyclonal antibody (Santa Cruz Biotechnology, Santa Cruz, CA, USA) diluted to 1:50 at 37 °C for 32 min. The signals were detected using the OptiView DAB IHC Detection Kit (Ventana Medical Systems, Tucson, AZ, USA). Counterstaining was performed using hematoxylin I (Ventana Medical Systems, Tucson, AZ, USA) for 4 min at room temperature.

### 2.9. IHC Scoring

Nuclear and/or cytoplasmic reactivity was assessed using the H-score, obtained by adding the products of the percentages of positively stained cells and their corresponding staining intensity (0, 1, 2, and 3). The ordinal values for staining intensity are as follows: 0 for no detectable staining, 1+ for weak reactivity that is mainly detectable at high magnification (20–40× objective), and 2+ or 3+ for intense staining reactivity (moderate and strong signal, respectively) that is easily detectable at low magnification (4× objective). Positivity was defined as an H-score of ≥100, the extent of staining ≥75%, or presence of 2+ or 3+ staining intensity.

### 2.10. Statistical Analyses

Data are expressed as mean ± standard deviation (SD) or as mean ± standard error of the mean (SEM). Groups were compared using Student’s *t*-test. Pearson coefficient was used when indicated. Progression-free survival (PFS) was calculated as the time from the date of diagnosis to disease progression, and overall survival (OS) was calculated as the time from diagnosis until death due to any cause. The PFS and OS curves were estimated using the Kaplan–Meier method.

## 3. Results

### 3.1. Expression of ID and E2A Genes in Ovarian Cancer

The analyses of ID and E2A from TCGA data are shown in Figure 1. The pan-cancer (PANCAN) normalized RNA sequencing (RNA seq; Illumina HiSeq, Illumina, Inc., San Diego, CA, USA) TCGA dataset (n = 7083) revealed decreased levels of ID 1–3 and increased levels of ID4 gene expression in ovarian cancer as compared to other cancers. In fact, *ID4* is particularly highly upregulated in ovarian cancer (Figure 1A). Cross-cancer alterations also revealed that ID 1–4 genes were amplified, and the E2A gene was deleted in ovarian cancer (Figure 1B,C). Moreover, the Genomic Identification of Significant Targets in Cancer (GISTIC) analysis was performed to identify the most significant regions of copy-number alterations in ovarian serous cystadenocarcinoma (*n* = 570). Interestingly, for ID-1, 58.1% of samples exhibited copy number gain, while 3.5% of samples portrayed a loss. For ID-2, 38.1% of samples portrayed a gain, and 18.9% exhibited a loss. For ID-3, we observed gain in 22.8% and loss in 43.3% of samples, and in ID-4, we detected gain in 50.7% and loss in 14.4% of samples. In contrast, E2A genes demonstrated copy number loss in 88.2% of samples but a gain in only 3.5% of samples. The ratio of gain in ID 1–4 genes to loss in *E2A* gene was 97.2% (554/570), with the majority of ovarian cancers exhibiting an ID gain or E2A loss (Figure 1D, Table 1). Analysis of the relationship between copy number alterations in the ID gene and corresponding mRNA expression revealed that ID-1 mRNA expression decreased in cases of ID-1 gene loss. However, in cases of ID-1 gain or amplification, the level of ID-1 mRNA expression was only slightly higher than that in cases of diploid ID-1 occurrence. In the majority of the cases, the level of ID-2 mRNA expression increased slightly, while that of ID-3 mRNA expression remained constant. However, the level of ID-4 mRNA expression increased in most cases regardless of the change in its copy number (Figure 1E). Additionally, we discovered that changes in copy numbers of ID and E2A were not related to the stage of ovarian cancer. However, the evaluation of changes in ID and E2A mRNA expression levels between early (stage 1) and advanced-stage (stage 3C, 4) ovarian cancers revealed that E2A mRNA expression increased in the early stages of ovarian cancer (Figure 1F).

### 3.2. Methylation Levels of ID and E2A Genes in Different Stages of Ovarian Cancer

We observed that the ID gene promoter was hypomethylated, while the E2A gene promoter was hypermethylated in ovarian cancer, regardless of the cancer stage (Figure 2A). In fact, analysis of the beta scores of ID and TCF3 genes, as determined by Human Methylation 27k (HM27) in TCGA data (*n* = 570), revealed that ID 1–4 genes were hypomethylated, while the E2A gene was hypermethylated (Figure 2B). Moreover, analysis of the relationship between ID methylation and mRNA expression revealed that although ID 1–4 were hypomethylated, only ID-2 and ID-4 mRNA expressions were increased, while ID-1 and ID-3 levels were not (Figure 2C). In contrast, E2A was hypermethylated in most cases, and its mRNA expression level was also decreased (Figure 2D). Furthermore, the E2A mRNA expression level further decreased with an increase in the E2A copy number loss, and the expression level of E2A mRNA in ovarian cancer was always low, irrespective of the expression level of ID mRNA (Figure 2D).

### 3.3. Association of ID and E2A alterations with Overall Survival

OncoPrint analysis of ID 1–4 genes revealed that 28% (161/575) of ovarian cancers demonstrated an alteration in these genes (Appendix A). Most of these alterations included an increase in the copy number of a particular gene, followed by an increase in its mRNA expression level. Furthermore, an evaluation of the associations between ID and/or E2A alterations and the OS of patients revealed that patients with ID-3 gene alterations demonstrated poor OS (*p* < 0.05) (Figure 2E).

### 3.4. Clinical Validity of ID and E2A Alterations in Ovarian Cancer Patients

To evaluate the clinical validity of ID and E2A alterations, we performed qRT-PCR and IHC analyses in high-grade serous ovarian carcinoma (HGSOC) samples from patient with an advanced stage of the disease (stages III–IV). Incidentally, qRT-PCR analysis revealed that the ID genes were upregulated and E2A was downregulated in the ovarian tissues of ovarian cancer patients, compared to those without ovarian cancer (Figure 3A,B). In fact, in ovarian cancer, the mRNA expressions of ID-1, ID-3, and ID-4 increased with a 25.13-fold (*p* < 0.001), 4.00-fold (*p* < 0.01), and 14.31-fold change (*p* < 0.001), respectively (Figure 3A). We also analyzed the ratio of mRNA expressions of ID 1–4 to E2A in ovarian cancer and normal ovarian tissues (Figure 3B). We observed that ID 1–4 had high expression levels in the ovarian cancer tissues than in normal ovarian tissues (for ID-1/E2A at *p* < 0.01, for ID-2/E2A at *p* < 0.01, for ID-3/E2A at *p* < 0.01, and for ID-4/E2A at *p* < 0.01).

Additionally, IHC analysis of cancerous (*n* = 23) and normal (*n* = 10) ovarian tissues revealed higher expression levels of ID-3 and ID-4 and lower expression levels of E2A in the cancerous ovarian tissues, compared to normal tissues. Interestingly, immunostaining did not reveal a detectable expression of ID-1 and ID-2 in the cancerous ovarian tissues (Figure 3C). The Kaplan–Meier curves for PFS and OS according to ID and E2A expressions are depicted in Figure 3D. Patients with low ID-3 expression levels showed better OS than those with high ID-3 expression (*p* = 0.042). However, the expression levels of ID-1, ID-2, ID-4, and E2A did not show a significant effect on survival outcomes.

### 3.5. Induction of the E2A Splice Variant E47 Causes Cell Growth Arrest in Ovarian Cancer Cells

To increase bHLH activity in aggressively growing human ovarian cancer cells, we stably transduced the human ovarian cancer cell line SCOV-3 with luciferase (Luc) and a tamoxifen-inducible form of E47 fused to an MER, thereby generating the SKOV-3/Luc/E47^MER^ cell line. These SKOV-3/Luc/E47^MER^ cell lines express high levels of E47 and exhibit nuclear localization of E47 that can be induced by tamoxifen treatment.

As shown in Figure 4A, E47^MER^-induced SKOV-3 cells underwent growth arrest or death when incubated with tamoxifen for 3 d. In contrast, the tamoxifen treatment of SKOV-3 cells lacking ectopic E47 expression did not result in cell growth arrest, thereby indicating that the cell cycle arrest was induced by E47 activity and not tamoxifen. Previous studies have reported that upon induction of E47 activity in PDA cells, Ki67 expression levels are diminished [15,24]. Similarly, we observed that the induction of E47 activity led to a rapid decrease in Ki67 expression levels in ovarian cancer cells (Figure 4B). Moreover, we observed a decline in the number of tamoxifen-induced SKOV-3/Luc/E47^MER^ cells compared to uninduced SKOV-3/Luc cells (Figure 4C). In a previous study using PDA cells, E47 had induced G0/G1 arrest or cell death [15]. Therefore, we used flow cytometry to assess DNA content for determining the stage of the cell cycle, G0/G1, S, or G2/M, wherein E47 arrested ovarian cancer cell growth; interestingly, we observed that E47 expression led to G0/G1 arrest and apoptosis (Figure 4D). Furthermore, when E47 was induced with tamoxifen for 2 d, followed by an additional 2 d culture without tamoxifen, the extent of apoptosis was significantly increased in the tamoxifen-treated cells (Figure 4E).

### 3.6. E47 Induces Global Changes in the Expression Levels of Genes Associated with Cell Cycle, Cancer, and Tissue Differentiation

To evaluate the changes induced by E47 with respect to gene expression, we performed qRT-PCR analysis of tamoxifen-treated and untreated SKOV-3 cells (Figure 4F). Incidentally, E47 induced considerable changes in gene expression in the ovarian cancer cells. Particularly, ovarian cancer-associated cell cycle activators, namely cyclin A2 (*CCNA2*) and aurora kinase A (*AURKA*), were downregulated by E47, thereby revealing the conserved effects of E47 in ovarian cancer cells.

The functions of E47 and the genes significantly associated with E47 alteration were predicted by GO and KEGG pathway enrichment analyses of differentially expressed genes in DAVID (https://david.ncifcrf.gov/summary.jsp (accessed on 1 November 2019). The GO enrichment analysis predicted the functional roles of the target host genes based on three aspects: biological processes, molecular functions, and functional annotation tool; the results revealed that E47 induction in ovarian cancer affected the genes involved in defense response, platelet-derived growth factor binding, and extracellular matrix. Additionally, KEGG analysis revealed that 10 pathways, including cell cycle, DNA replication, cancer-associated pathways, and the p53 signaling pathway, were related to E47 functions in ovarian cancer (Appendix A).

### 3.7. E47 Inhibits Ovarian Cancer Formation In Vivo

The induction of E47 activity was sufficient to inhibit tumor growth in ovarian cancer cells in vitro, therefore we hypothesized that altering the bHLH transcription networks might also inhibit tumor growth of ovarian cancer cells in vivo. To evaluate the effects of E47 on tumor growth dynamics, we treated control SKOV-3/Luc cells and SKOV-3/Luc/E47^MER^ cells with implanted tamoxifen pellets from d 3 to d 45 after cell implantation into SCID/beige mice. As shown in Figure 4A, during the 77 d treatment course, mice with tamoxifen-treated control SKOV-3/Luc cells produced large tumors, whereas those with tamoxifen-induced SKOV-3/Luc/E47^MER^ cells produced tumors that were significantly reduced in size (*p* = 0.00012) (Figure 5A,B).

## 4. Discussion

Even though several novel anti-cancer drugs have been proposed to treat ovarian cancer, the majority of the currently available targeted therapeutic agents have limited efficacy. Therefore, we analyzed ID and E2A as potential candidates for targeted therapy in ovarian cancer patients. In our study, TCGA data analysis revealed that the majority of patients with serous ovarian cancer exhibited copy number gain in ID-1 and ID-4, while 89.3% of them showed copy number loss in E2A. Furthermore, ID 1–4 genes were hypomethylated, but E2A was hypermethylated, and in ovarian cancer the expression level of ID-4 mRNA increased while that of E2A mRNA decreased. Incidentally, the decreased expression of E2A mRNA was highly prominent in the advanced stages of the disease. Moreover, ID expression level increased and E2A expression level decreased in tumor tissues of HGSOC patients. Interestingly, low expression levels of ID-3 were consistent with better patient OS.

The E2A is a ubiquitously expressed transcription regulator, and the E2A gene encodes two bHLH transcription factors, E12 and E47 [25], which are characterized by their broad expression patterns and DNA-binding abilities [26,27]. The role of E2A in tumor growth is often overlooked, because its loss of function is not always reflected in terms of expression levels detected in microarray or RNA seq studies. However, high levels of ID proteins can inhibit the activity of E2A. Incidentally, E47 plays a critical role in promoting B-cell lymphopoiesis, T-cell development, myogenesis, and cell proliferation [15,23,28,29]. However, its role in ovarian cancer is unclear. Reportedly, ID proteins are transcriptional regulators that play critical roles in normal cell growth and differentiation [30]. The primary function of ID 1–4 proteins is to bind to bHLH transcription factors and inhibit their activities [31]. The bHLH proteins activate transcription by forming heterodimers that bind to regulatory enhancer box sequences in target genes. Since ID proteins lack a basic DNA-binding domain, they can function as dominant-negative regulators of bHLH proteins by forming ID/bHLH heterodimers. In fact, they have significant roles in a variety of biological processes that regulate tumorigenesis, such as G1/S cell cycle transition [32], activation of potential proto-oncogenes [33], and exogenous tumor cell growth and metastasis [9].

In this study, we demonstrated that ID-3 is highly expressed in HGSOC, and is negatively associated with OS. A potential explanation of this phenomenon is that ID-3 promotes Wnt/β-catenin signaling, which when activated plays a significant role in HGSOC chemotherapy resistance. In fact, Huang et al. [34] have established that ID-3 promotes the stemness of intrahepatic cholangiocarcinoma, which in turn leads to the activation of Wnt/β-catenin signaling. Moreover, numerous cancers have hyperactive β-catenin signaling, and this pathway is associated with cancer initiation, progression, therapeutic resistance, and recurrence [35,36]. Nagaraj et al. [37] have reported that the Wnt/β-catenin pathway maintains the stemness of HGSOC and platinum resistance, thereby indicating its association with intrinsic and acquired platinum chemo-resistance. Furthermore, the Wnt/β-catenin pathway is associated with suppression of anti-cancer immune responses within tumor microenvironment [38,39] and enhancement of tumor angiogenesis [40]. Therefore, we believe that high expression of ID-3 may influence survival outcomes in patients with HGSOC.

We hypothesized that dysregulated ID protein activity could be integral to ovarian cancer pathogenesis; therefore, modulating the bHLH/ID balance could be a promising approach for treating ovarian cancer. We discovered an imbalance of ID and E2A in ovarian cancer cells and tissues of ovarian cancer patients, and realized that a change in bHLH might affect their survival outcomes. Furthermore, we demonstrated that exogenous E47 expression induced cell death in vitro and inhibited tumor growth in the SKOV-3 mouse model. Notably, temporary induction of E47 in vitro and in vivo produced G0/G1 arrest and apoptosis. Moreover, the discovery that E47 can induce genes that are normally responsive to the tumor suppressor p53 is particularly interesting because SKOV-3 cells do not express p53. Hence, E47 is sufficient to induce genes downstream of p53, even in the absence of p53, as demonstrated for the cyclin-dependent kinase inhibitor p21 (CDKN1A) [41]. We also proved that the restoration of bHLH expression in ovarian cancer can be achieved by inducing E47 activity, thereby suggesting that any strategy to promote E47 activity can offer a promising approach for treating ovarian cancer, similar to research performed for pancreatic cancer [42].

## 5. Conclusions

In conclusion, this study revealed an imbalance in the ID and E2A expressions in ovarian cancer cells and tumor tissues, with particular emphasis on the correlation between ID-3 expression and poor OS. Therefore, E2A (E47) overexpression may induce ovarian cancer cell growth arrest by restoring the bHLH/ID balance.

## Figures and Tables

**Figure 1 cancers-14-02903-f001:**
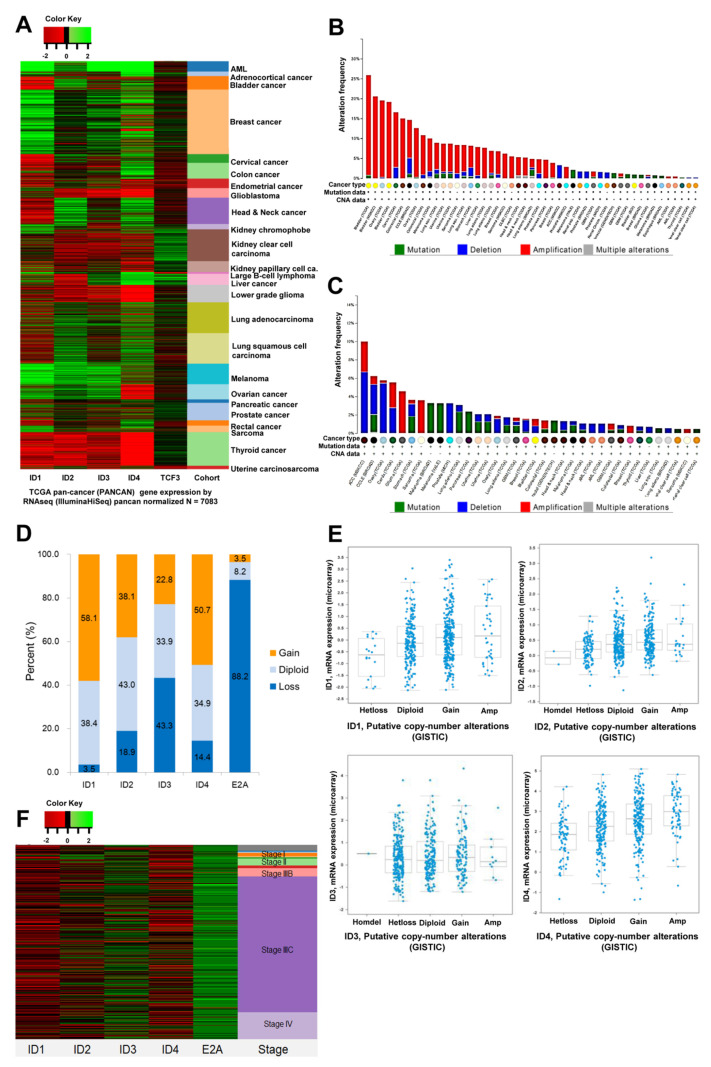
TCGA data analysis of gene alteration for ID and E2A. (**A**) TCGA pan-cancer (PANCAN) gene expression by RNAseq (IlluminaHiSeq) pancan normalized (*n* = 7083), (**B**) Cross-cancer alteration summary for ID1−4, (**C**) Cross-cancer alteration summary for E2A, (**D**) Putative copy-number alterations from GISTIC in ovarian serous cystadenocarcinoma, (TCGA, *n* = 570), (**E**) Analysis of the relationship between copy number of ID1–4 gene and mRNA expression in ovarian serous cystadenocarcinoma using cBioportal, (**F**) Copy number alterations from GISTIC according to FIGO stage in ovarian serous cystadenocarcinoma (TCGA, *n* = 570).

**Figure 2 cancers-14-02903-f002:**
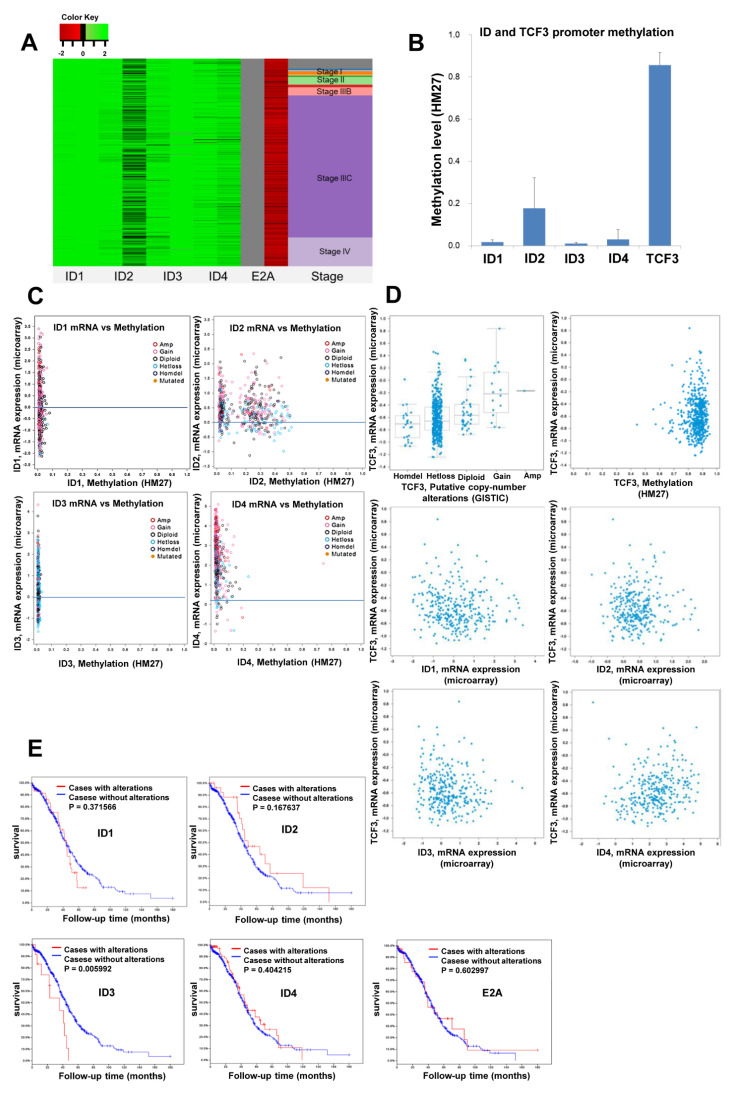
TCGA data analysis of methylation for ID and E2A. (**A**) DNA Methylation (HM27) in Ovarian Serous Cystadenocarcinoma according to stage (TCGA, *n* = 616). ID gene promoters werehypomethylated and the E2A gene promoter was hypermethylated regardless of stage, (**B**) Beta scores of ID and TCF3 genes determined by HM27 in TCGA data (*n* = 570). ID1-ID4 were hypomethylated and E2A was hypermethylated, (**C**) The relationship between methylation of ID gene and mRNA expression in ovarian serous cystadenocarcinoma using cBioportal, (**D**) Analysis of the relationship between copy number of E2A gene, ID1–4 gene and mRNA expression in ovarian serous cystadenocarcinoma using cBioportal. E2A was hypermethylated in most cases and mRNA expression levels decreased. E2A mRNA expression level decreased with increased E2A copy number loss. The expression level of E2A mRNA was low regardless of the mRNA expression level of the ID gene, (**E**) Overall survival in ovarian serous cystadenocarcinoma according to ID and E2A gene alterations. ID-3 gene alterations showed poor overall survival (*p* < 0.05).

**Figure 3 cancers-14-02903-f003:**
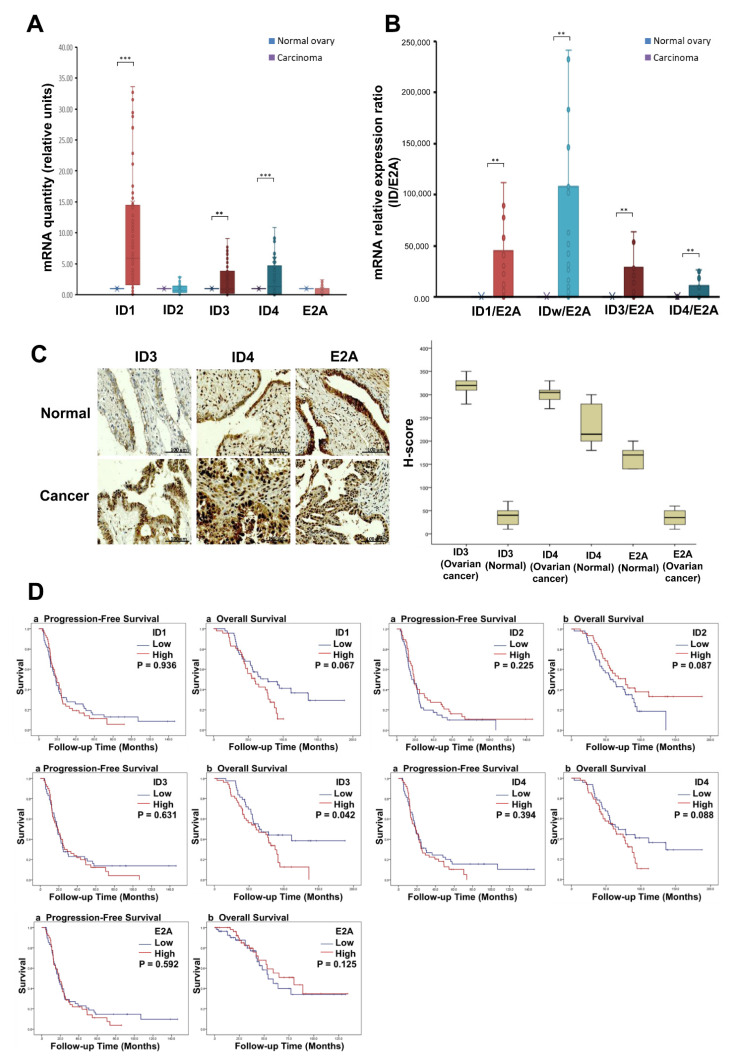
ID, E2A expression and survival outcomes in patients with ovarian cancer (*n* = 103). (**A**) E47 induces expression of ID1, ID3, ID4 and E2A target genes in ovarian cancer tissue and normal ovarian tissue. The mRNA expression of ID1, ID3 and ID4 in ovarian cancer increases with a change of 25.13 (*p* < 0.001), 4.00 (*p* < 0.01) and 14.31-fold (*p* < 0.001) respectively. (**B**) Expression ratio of ID1–4/E2A target genes in ovarian cancer tissue and normal ovarian tissue. ID1–4 had higher expression levels in ovarian cancer tissue than normal ovarian tissue (for ID1/E2A at *p* < 0.01, for ID2/E2A at *p* < 0.01, for ID3/E2A at *p* < 0.01, for ID4/E2A at *p* < 0.01). ID genes were all significantly upregulated in ovarian cancer tissue. (**C**) Representative microscopic images (200× magnification, scale bar 100 μm) of ID and E2A and immunohistochemistry score in ovarian cancer. The expression levels of ID-3, ID-4 were higher in cancerous ovarian tissues than in normal ovarian tissues, and E2A was lower (normal group: *n* = 10, cancer group: *n* = 23). (**D**) Kaplan–Meier curves of progression-free survival and overall survival according to ID-1–4 and E2A gene expression in ovarian cancer patients. Patients with low ID-3 expression levels showed better overall survival than those with high ID-3 expression levels(*p* = 0.042). ** *p* < 0.01, *** *p* < 0.001).

**Figure 4 cancers-14-02903-f004:**
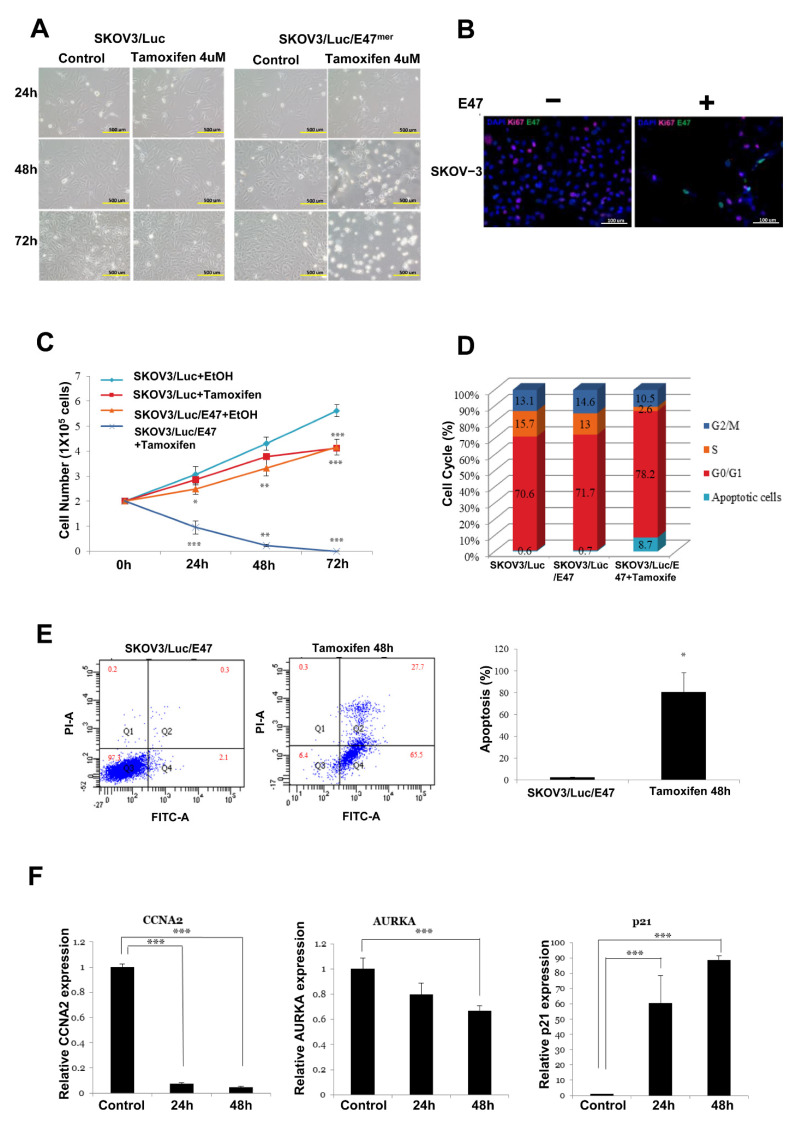
E47 induces cell growth arrest or cell death in SKOV-3 cells. (**A**) Microscopy (scale bar, 500 μm) showed that E47^mer^ induced SKOV-3 cell growth arrest when cells were incubated with tamoxifen. (**B**) Immunostaining (scale bar, 100 μm) for the replication markers Ki67 (red) and DAPI (blue), ×200. (**C**) Growth curves (log scale) for SKOV-3/Luc and SKOV-3/Luc/E47^mer^ cells. (**D**) Percentage of cells in individual cell cycle phases determined by flow cytometry. (**E**) Analysis of apoptosis following tamoxifen treatment using flow cytometry in SKOV-3/Luc/E47^mer^ cells. (**F**) qRT-PCR for CCNA2, AURKA, P21, TOP2A transcripts in all SKOV3/E47 lines. (* *p* < 0.05, ** *p* < 0.01, *** *p* < 0.001, Q1: necrotic cell, Q2: late stage apoptotic cells or necrotic cells, Q3: live cells, Q4: early apoptotic cells).

**Figure 5 cancers-14-02903-f005:**
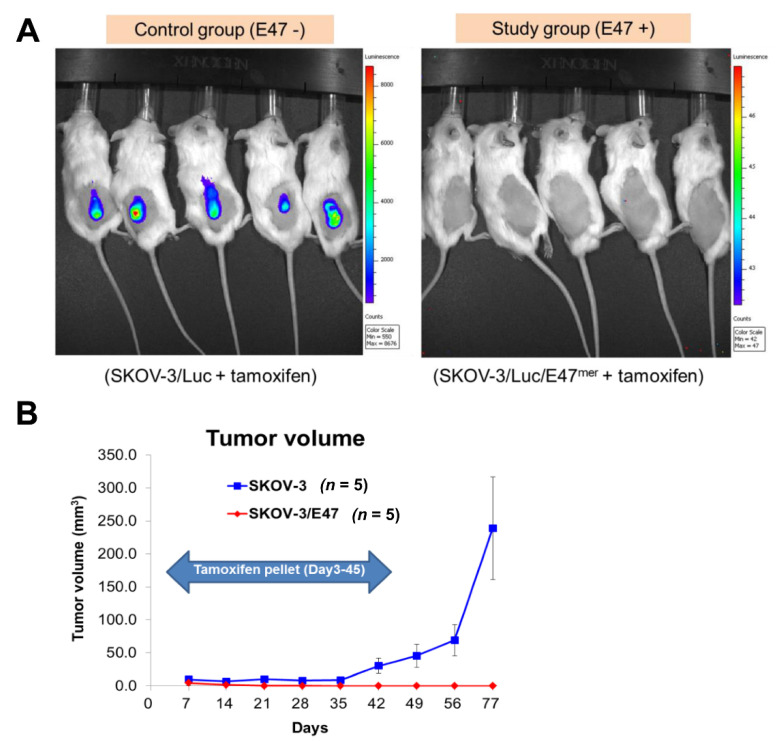
E47 inhibits ovarian cancer cell tumorigenesis in vivo. (**A**) Bioluminescence imaging on day 77 using Xenogen IVIS Imaging System (control group: E47 −, *n* = 5 vs. study group: E47 +, *n* = 5). (**B**) E47-inhibited tumor growth in SKOV-3 cells transplanted NSG mouse.

**Table 1 cancers-14-02903-t001:** Putative copy number alterations from GISTIC in Ovarian Serous Cystadenocarcinoma, (TCGA, *n* = 570).

Copy Number	Gene (No, %)				
	ID1	ID2	ID3	ID4	E2A
−2	0 (0.0)	2 (0.4)	1 (0.2)	0 (0.0)	26 (4.6)
−1	20 (3.5)	106 (18.6)	246 (43.2)	82 (14.4)	477 (83.7)
0	219 (38.4)	245 (43.0)	193 (33.9)	199 (34.9)	47 (8.2)
1	289 (50.7)	195 (34.2)	118 (20.7)	223 (39.1)	18 (3.2)
2	42 (7.4)	22 (3.9)	12 (2.1)	66 (11.6)	2 (0.4)

## Data Availability

The raw data supporting the conclusions of this article will be made available by the authors without undue reservation.

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
