# Peer review of "Expression Profiles of ID and E2A in Ovarian Cancer and Suppression of Ovarian Cancer by the E2A Isoform E47"

_cancers, 2022, doi:10.3390/cancers14122903_

Round 1

Reviewer 1 Report

The  authors improved the revised version of the manuscript addressing all requests.

This version  can be considered suitable for publication.

Author Response

: We would like to express our gratitude to the editor and the reviewers for taking the time to carefully read and consider the report on our present study. We appreciate the opportunity to revise the manuscript by addressing the reviewers’ comments based on their constructive guidance to help you and the reviewers in your final decision.

Reviewer 2 Report

The authors have responded to the majority of the comments. However, a few concerns must be addressed.

  1. They did not imply how much volume was injected and what solvent was used to deliver Tamoxifen.
  2. In an apoptosis assay, what does the quadrant represent?
  3. The authors were asked for the expression of cell cycle protein. However, it was not included in the revised paper by the authors.
  4. The mouse tumor tissue should be examined for an IHC investigation.

Author Response

Reviewer 2

The authors have responded to the majority of the comments. However, a few concerns must be addressed.

: We appreciate this positive statement regarding the overall evaluation. Following the constructive comments and suggestions, we have revised the entire manuscript and provide the point-by-point responses as follows.

1.They did not imply how much volume was injected and what solvent was used to deliver Tamoxifen.

Answer: Thank you for your comments. Tamoxifen was delivered to the mouse using tamoxifen pellets (25mg/pellet) that was purchased from Innovative Research of America (Sarasota, Florida, USA). The picture below is tamoxifen pellet used in our study and the trocar used for subcutaneous implantation. Implanted 25mg tamoxifen pellet integrates the three principles of diffusion, erosion, and concentration gradients. It generates a finished pellet with a biodegradable matrix that effectively and continuously releases the active product in the animal. The pellet delivery system can help the investigator take charge of standardizing and reproducing research results.

2.In an apoptosis assay, what does the quadrant represent?

Answer: Thank you for your nice comments. In the apoptosis assay, Q1 represents necrotic cells, Q2 represents late stage apoptotic cells or necrotic cells, Q3 represents live cells, Q4 represents early apoptotic cells. Following the comment, we revised the manuscript to present more details about the apoptosis assay. Figure 4E is a comparison of flow cytometry results after 48 hours of tamoxifen treatment in SKOV-3/Luc/E47mer cells in our study. After tamoxifen treatment, the proportion of PI positive cells (Q2 and Q4) increased, confirming that apoptosis occurred.

We added the following sentences in the Figure 4 legends (Line 355 in revised manuscript)

(Q1: necrotic cell, Q2: late stage apoptotic cells or necrotic cells, Q3: live cells, Q4: early apoptotic cells)

3.The authors were asked for the expression of cell cycle protein. However, it was not included in the revised paper by the authors.

Answer: Thank you for pointing out the need of clarification. We evaluated the changes induced by E47 in the expression levels of genes. We performed qRT-PCR analysis in tamoxifen-treated and tamoxifen-untreated SKOV-3 cells. In the gene expression level analysis, it showed that the cell cycle activators were significantly downregulated by E47, and in the KEGG pathway enrichment analysis, it also showed that the expression of the cell cycle pathways were related to the E47 function in ovarian cancer cell. It would have been better if the cell cycle protein was additionally confirmed through Western blot or immunostaining. We did not further analyze the expression of cell cycle protein including CCNA2 and AURKA through Western blot or immunostaining as the analysis results of gene expression level by qRT-PCR showed a significant difference.

4.The mouse tumor tissue should be examined for an IHC investigation.

Answer: Thank you for your comments. Our in-vivo study was to confirm the inhibition of ovarian cancer cell tumorigenesis by E47. As the reviewer suggested, it may be meaningful to compare mouse tumor tissue of the tamoxifen-treated group and the untreated group by performing IHC. However, it was impossible to compare the two groups because tumors did not grow in the tamoxifen-treated group. The tumor was not formed in the mouse that was treated with tamoxifen.

This manuscript is a resubmission of an earlier submission. The following is a list of the peer review reports and author responses from that submission.

Round 1

Reviewer 1 Report

In their manuscript Lee et al. analyze the expression pattern of ID and E2A in ovarian cancer. The study reports that patients bearing high ID3 levels show a worse survival , while low E2A levels were found in cancerous ovarian tissues than normal ones. Interestingly, E2A induction in SKOV-3/Luc cells transduced with tamoxifen inducible E47, an E2A splice variant,showed a strong antitumorali action in both in vitro and vivo assays.

The study sounds to be original and identify new possible molecular targets for ovarian cancer. However, before to consider this paper suitable for publication:

  1. The authors should better introduce ID and E2A factors in the abstract.
  2. The authors should improve the description of methods used. Please, describe how cell cycle analyses was performed.
  3. A description of the meaning of apoptotic cell death should be included in the text. Which kit was used by authors for the analyses? Specify vendor and procedure in the text and, describe these aspects in method section. Figure on apoptotic cell death reports PI on y axis and FITC on x axis. Is it referred to annexin V? It is not clear. Please, clarify this aspect.
  4. The paper shoul be carefully read by a mother tongue speaker.

Reviewer 2 Report

One of the deadliest gynecological cancers in women is ovarian cancer; even after achieving complete remission following the first treatment, most patients with advanced-stage cancer return. Several new medicines have been proposed to improve survival outcomes; however, their efficacy in ovarian cancer has been restricted. It has long been believed that ID proteins could be used as potential therapeutic targets. The degree of ID-1 expression is linked to ovarian cancer's malignant potential, resulting in unfavorable prognoses. In ovarian cancer cells, inhibiting ID proteins causes cell-cycle arrest and apoptosis. However, to develop more effective treatment ideas, a detailed understanding of ovarian cancer in terms of ID and E2A expression levels is required. In this study, the authors investigated the degree of mutation or expression of ID1-4 and E2A genes and the significance of E47 as a growth suppressor in ovarian cancer cells using ovarian cancer samples from the TCGA database. They also determined the function of overexpressing the E2A splice variant E47 in ovarian cancer patients and demonstrated ID and E2A expression in tumor tissues of ovarian cancer patients. However, I have a few concerns from this study.

  1. What was the rationale behind choosing the SKOV3 cell line?
  2. Line 141; Subsequently, 3,000,000 μL and 300 μL of SKOV-3/Luc and SKOV-3/Luc/E47MER cells, respectively, were subcutaneously implanted into SCID/beige mice. What exactly do you mean when you say 3,000,000 μL?
  3. Tamoxifen pellets (25 mg) were implanted subcutaneously at other sites to analyze the effect of E47 on tumorigenesis and the growth of SKOV-3/Luc cells. However, it is unclear how the pellet was administered in terms of volume and solvent usage.
  4. The ethical committee approval for animal studies is not addressed in this article.
  5. Several experiment methods are missing, such as IF, apoptosis, etc. 
  6. Initially, the authors indicated that 14 mice were used in the technique. However, no data showed more than ten mice in the study (Figure 5A). Further, the authors compare the tumor volume of SKOV-3 and SKOV-3/E47 in figure 5B. Next, the authors should consider the control mice with SKOV3/Luc and no tamoxifen while comparing the results. Overall, the animal study is unclear and looks to be deceptive. 
  7. How did the mouse euthanize when the experiment was completed?
  8. What were the patient sample's exclusion and inclusion criteria?
  9. What do the quadrants in apoptosis picture 4E represent? It's also unclear whether the authors carried it out.
  10. In picture 4A, the scale bar does not clear. What scale bar was used in 4B and should be included as well?
  11. For cell cycle checkpoints, western blots protein expression should be examined.
  12. The current version of the text does not accurately reflect the proposed research. More research, both in vitro and in vivo, is needed to prove the hypothesis.